# Diagnostic Accuracy of Various Immunochromatographic Tests for NS1 Antigen and IgM Antibodies Detection in Acute Dengue Virus Infection

**DOI:** 10.3390/ijerph19148756

**Published:** 2022-07-19

**Authors:** Mughees Haider, Saira Yousaf, Asifa Zaib, Azza Sarfraz, Zouina Sarfraz, Ivan Cherrez-Ojeda

**Affiliations:** 1Research, Sargodha Medical College, University of Sargodha, Sargodha 40100, Pakistan; mugheeskhokhar89@gmail.com (M.H.); sairayousaf07@gmail.com (S.Y.); 2Research, Punjab Medical College, Faisalabad Medical University, Faisalabad 38000, Pakistan; asifazaib15@gmail.com; 3Pediatrics and Child Health, Aga Khan University, Karachi 74000, Pakistan; azza.sarfraz@aku.edu; 4Research and Publications, Fatima Jinnah Medical University, Lahore 54000, Pakistan; 5Allergy, Immunology and Pulmonology, Universidad de Especialidades Espíritu Santo, Samborondón 0901-952, Ecuador

**Keywords:** infectious disease control, immunochromatographic tests, dengue, point-of-care, public health

## Abstract

Introduction: Rapid diagnostic tests (RDTs) were evaluated, in this paper, for their utility as a reliable test, using resource-constrained studies. In most studies, NS1 antigen and immunoglobulin M (IgM)-based immunochromatographic tests (ICTs) were considered for acute phase detection. We aimed to evaluate the diagnostic accuracy of NS1, IgM, and NS1/IgM-based ICTs to detect acute dengue virus (DENV) infection in dengue-endemic regions. Methods: Studies were electronically identified using the following databases: MEDLINE, Embase, Cochrane Library, Web of Science, and CINAHL Plus. Keywords including dengue, rapid diagnostic test, immunochromatography, sensitivity, specificity, and diagnosis were applied across databases. In total, 15 studies were included. Quality assessment of the included studies was performed using the QUADAS-2 tool. All statistical analyses were conducted using RevMan, MedCalc, and SPSS software. Results: The studies revealed a total of 4135 individuals, originating largely from the Americas and Asia. The prevalence of DENV cases was 53.8%. Pooled sensitivities vs. specificities for NS1 (only), IgM (only) and combined NS1/IgM were 70.97% vs. 94.73%, 40.32% vs. 93.01%, and 78.62% vs. 88.47%, respectively. Diagnostic odds ratio (DOR) of DENV for NS1 ICTs was 43.95 (95% CI: 36.61–52.78), for IgM only ICTs was 8.99 (95% CI: 7.25–11.16), and for NS1/IgM ICTs was 28.22 (95% CI: 24.18–32.95). ELISA ICTs yielded a DOR of 21.36, 95% CI: 17.08–26.741. RT-PCR had a DOR of 40.43, 95% CI: 23.3–71.2. Heterogeneity tests for subgroup analysis by ICT manufacturers for NS1 ICTs revealed an χ^2^ finding of 158.818 (df = 8), *p* < 0.001, whereas for IgM ICTs, the χ^2^ finding was 21.698 (df = 5), *p* < 0.001. Conclusion: NS1-based ICTs had the highest diagnostic accuracy in acute phases of DENV infection. Certain factors influenced the pooled sensitivity, including ICT manufacturers, nature of the infection, reference method (RT-PCR), and serotypes. Prospective studies may examine the best strategy for incorporating ICTs for dengue diagnosis.

## 1. Introduction

Dengue is a flavivirus infection spread by *Aedes aegypti* and *Aedes albopictus* mosquitoes, with four antigenically distinct dengue viruses (DENVs, serotypes 1–4) causing infection, and is a significant public health problem [1]. It has rapidly spread to nearly half the world’s population and has caused epidemics in these regions with continued geographical expansion [2]. It has caused 400 million annual infections, which have risen exponentially over the last few decades [3]. Dengue virus and antigen detection are the most accurate diagnostic tools during the first five days of illness, i.e., the period of viremia, as IgG and IgM antibodies are not produced until 5–7 days after the onset of symptoms in primary infections [4].

The methods currently used to detect acute DENV infections that are endorsed by the World Health Organization are isolation of dengue viral antigens and detection of viral nucleic acid in blood categorized by a positive reverse transcriptase–polymerase chain reaction (RT-PCR), immunoglobulin type M (IgM) seroconversion, and/or a four-fold or greater rise in immunoglobulin G (IgG) antibody titers in paired blood samples collected at least 14 days apart [5,6]. A reverse transcriptase–polymerase chain reaction (RT-PCR) assay can more accurately confirm the active infection and serotype of the dengue infection [7]. The plaque reduction neutralization test (PRNT) identifies serotype-specific antibodies, but it is even more laborious and expensive than other methods and hence not routinely used [8]. Another diagnostic method is immunoglobulin type M (IgM) antibody capture enzyme-linked immunosorbent assay (MAC-ELISA), which is challenging to interpret as IgM remains elevated for 2–3 months after infection [9]. The NS1 capture ELISA was developed following reports of high NS1 antigen titers in the acute phase of the disease [10]. All serological assays can exhibit some degree of cross-reactivity with other flaviviruses such as Zika, Japanese encephalitis, and yellow fever viruses [11].

A diagnostic tool gaining prominence is rapid diagnostic tests (RDTs) which is a convenient option, particularly in resource-constrained and dengue-endemic countries with limited capability to conduct RT-PCR or ELISA [12]. RCTs typically detect dengue virus nonstructural protein 1 (NS1) antigen, IgM, IgG, and IgA antibodies with higher specificity (~90%) than sensitivity (~10–99%) [13,14,15,16,17,18]. Although RDTs are not as sensitive as PCR or ELISA, they are quick, convenient, and require no expertise. Their ability to rapidly diagnose dengue virus (DENV) infection in communities and clinical settings is an attractive option for resource-constrained settings [19]. The World Health Organization recommends coordinated care at the primary healthcare level as most DENV-infected patients are treated in these units and require testing that may be performed without laboratories in proximity [5]. Here, we report the clinical sensitivity and specificity of different immunochromatographic tests (ICTs) detecting NS1 antigen, IgM antibodies, and combined NS1/IgM detection in acute DENV infection. To our best knowledge, this is the first study to compare multiple ICTs and their performance against different reference tests, serology types, and acute vs. primary infection for detection of active DENV infection.

## 2. Materials and Methods

### 2.1. Search Strategy

Searches were conducted on the following databases: MEDLINE (1966—4 May 2022), EMBASE (1994—4 May 2022), Cochrane Library, Web of Science, and CINAHL Plus. The search terms used were ‘dengue’, ‘rapid diagnostic test’, ‘immunochromatography’, ‘sensitivity’, ‘specificity’, and ‘diagnosis’, which were searched for each country identified as endemic by the Centers for Disease Control and Prevention [20]. Reference lists of each of the selected articles were hand-searched for additional studies (umbrella framework) [21]. No language restrictions were imposed. The protocol was registered with PROSPERO 2022: CRD42022334839).

### 2.2. Study Selection Using Standardized Quality Assessment Criteria

Abstracts of identified studies were printed and, if potentially relevant, were obtained as full-text articles. Two authors (ZS and AS) conducted the quality assessment of included studies utilizing the QUADAS-2 assessment tool; this tool was solely created for diagnostic test accuracy reviews [22]. Studies were considered to be of high methodological quality if they had a low concern and low risk of bias. The findings were presented in a study-by-study graph under risk of bias and applicability concerns figure legends.

Studies were excluded if they had any of the following characteristics: (1) use of inappropriate reference assays to assign true positive/true negative status to study samples, including ‘in-house’ assays for which the diagnostic accuracy had not been previously established; (2) inappropriate study population (such as convalescent samples only); (3) the study was limited to the detection of IgG rather than IgM and IgG; (4) the number of study samples was insufficient; (5) incomplete description of samples, such that it was impossible to determine the timing of sample collection; (6) errors or inconsistencies in the published study data; (7) the exclusion of indeterminate results; (8) partial verification of the study samples or the use of multiple reference assays; or (9) the assay took more than 60 min to perform, such as immunoblot (IBT)-style assays. The list of excluded full-text studies is given in Appendix A.

### 2.3. Data Extraction

Two authors (ZS and AS) extracted data into a shared spreadsheet, with the final author (ICO) present for any disagreements. The data were extracted as “author/year/country”, region, DENV positive individuals (n), cohort size (N), prevalence (n/N, %), reference method, sample type, days post fever onset, ICT manufacturer, sensitivity (95% CI), specificity (95% CI), primary infection sensitivity (%), secondary infection sensitivity (%), and serotypes sensitivity (%).

### 2.4. Statistical Analysis

The ‘gold standard’ (or reference) assay was compared with the index test to define true positive (TP), false positive (FP), false negative (FN), and true negative (TN) values. The measures of diagnostic accuracy, sensitivity (SN), specificity (SP), positive likelihood ratio (LR+), negative likelihood ratio (LR−), positive predictive value (PPV), negative predictive value (NPV), Fisher Exact P-values, and diagnostic odds ratio (DOR) were then computed (Habbema et al., 2002). Individual study results were pooled to generate an overall estimate of diagnostic accuracy. Chi-square and I^2^ (Higgins et al., 2003) statistics were calculated before pooling to detect significant heterogeneity between subgroups. NS1 (only), IgM (only), NS1/IgM subgroup analysis in the acute phase DOR, Error Odds Ratio, Phi coefficient (to measure the strength of the relationship), and relative improvement over chance (RIOC, to measure predictive efficiency) were additionally computed. Utilizing the 2-by-2 table data, the Fisher Exact P-values and other relevant indices including an analysis of risk factors for analysis of the effectiveness of a diagnostic criterion for dengue based on multiple parameters (SN, SP, PPV, NPV, DOR, and error odds ratios), and other measures of association (Phi coefficient), were computed. The confidence intervals for the estimated parameters were computed by a general method as listed by Fleiss and colleagues (1981) [23]. The confidence intervals for RIOC and Phi coefficient as measures of predictive efficiency and strength of association in 2-by-2 tables were based on Farington and Loeber (1989) [24]. Definitions of listed statistical tests are enlisted in Table 1. A chi-square result of *p* < 0.1 was considered significant, given the low power of the test. I^2^ values had a continuous scale of 0–100%, with 0% defining no heterogeneity and 25, 50, and 75% tentatively assigned as limits of low, medium, and high heterogeneity (Higgins et al., 2003). If heterogeneity was not significant, a Mantel–Haenszel fixed-effects model (Mantel and Haenszel, 1959) was used to calculate results, and, when significant, a random-effects model was used (DerSimonian and Laird, 1986). Summary receiver operator characteristics (SROC) (Littenberg and Moses, 1993) were also calculated to give a final area under the curve (AUC) value for pooled and subgroup analyses. A summary ROC curve is a plot of the combined SN on the y-axis against (1-SP) on the *x*-axis. The 45° diagonal line connecting (0, 0) to (1, 1) is the ROC curve corresponding to random chance. The ROC curve for the gold standard is the line connecting (0,0) to (0,1) and (0,1) to (1,1). Therefore, the summary lines that were closest to the upper-left corner of the plot were considered nearest to the gold standard dengue testing format. Analyses were performed using RevMan 5.4.1, MedCalc Software (v 20.104), and IBM^®^ SPSS^®^ software (v. 27).

## 3. Results

In total, 1652 studies were identified by electronic searches. Abstracts were read, and 46 studies were retained for full-text quality assessment. Five studies were identified by reading reference lists and hand-searching journals (umbrella review search). In total, 51 studies were selected for full-text review against the inclusion and exclusion criteria. Only 15 studies were included according to the selection criteria, whereas 36 were excluded (Appendix A). Figure 1 shows a flow chart of the selection procedure.

### 3.1. Quality of Included Studies

The risk of bias and applicability concerns summary for every included study is depicted in Figure 2. Firstly, on assessing the risk of bias, nine studies had unclear risk and six had a low risk of bias for patient selection. The index test had an unclear risk of bias in six studies, whereas nine studies had a low risk of bias. On noting reference standards, all 15 studies had a low risk of bias. Flow and timing assessment revealed eight studies with unclear risk of bias, five studies with low risk of bias, and two with a high risk of bias. Secondly, the applicability concerns assessment revealed that 14 studies had a low risk of bias whereas only one had an unclear risk of bias. Index test assessment yielded low risks of bias for all 15 studies. Finally, the reference standard assessment determined that 14 studies had a low risk of bias and only one had an unclear risk.

### 3.2. Narrative Review of Included Studies

A summary of all included data is shown in Table 2. The overall prevalence of DENV cases was 53.8% in our entire sample. Reference methods used were viral isolation, NS1 capture ELISA, IgM capture ELISA, IgM seroconversion, IgM antibody capture enzyme-linked immunosorbent assay (MAC-ELISA), RT-PCR, and 4-fold increased titers on hemagglutination inhibition test (HAI). Three studies were from the Americas including Brazil, Colombia, and Peru, and 12 studies were from Asia including Cambodia, India, Myanmar, Taiwan, Thailand, and Vietnam. The total sample consisted of 4135 individuals with suspected DENV infection, of whom 2225 were detected as DENV positive through reference assays. Nine different ICTs detected NS1 antigen, including: (1) Dengue NS1 Ag STRIP™ (Biorad Laboratories, Marnes-La-Coquette, France), (2) SD Bioline Dengue NS1 Ag Rapid Test (Alere, North Chicago, IL, USA), (3) Dengue NS1 Detect Rapid Test (1st generation) (InBios International, Seattle, WA, USA), (4) Panbio™ Dengue Early Rapid, (5) J. Mitra Dengue Day 1 Test, (6) Dengue Ag Rapid Test-Cassette (CTK Biotech, Inc., Poway, CA, USA), (7) CareUS Dengue Combo (WellsBio, Seoul, Korea), (8) Humasis Dengue Combo (Humasis, Anyang, Korea), and (9) Wondfo Dengue Combo, China. Five different ICTs detected IgM antibodies, including: (1) SD BIOLINE Dengue DUO^®^ (Standard Diagnostic Inc., Seoul, Korea), (2) Dengue Combo Rapid Test-Cassette (Chembio Diagnostics, Inc., Medford, NY, USA), (3) CareUS Dengue Combo (WellsBio, Seoul, Korea), (4) Humasis Dengue Combo (Humasis, Anyang, Korea), and (5) Wondfo Dengue Combo, China.

### 3.3. Individual and Pooled Study Diagnostic Accuracy Results

The individual studies using dengue ICT for dengue NS1 only (Pooled SN = 70.97%, SP = 94.73%), IgM only (Pooled SN = 40.32%, SP = 93.01%), and combined NS1/IgM (Pooled SN = 78.62%, SP = 88.47%) detection in acute studies were pooled and analyzed separately, as discussed in subsequent sections. The summary forest plot outcomes for all tests are depicted in Figure 3.

Figure 4 aims to provide a glance at the diagnostic odds ratio findings for: (i) NS1 (only) (DOR = 43.95, 95% CI = 36.61–52.78), (ii) IgM (only) (DOR = 8.99, 95% CI = 7.248–11.157), (iii) NS1/IgM (DOR = 28.22, 95% CI = 24.179–32.946), (iv) ELISA (DOR = 21.362, 95% CI = 17.08–26.741), and (v) RT-PCR (DOR = 40.432, 95% CI = 23.297–71.211). The findings are further elaborated in subsequent sections.

### 3.4. Subgroup Analysis by NS1 (Only) in the Acute Phase

The diagnostic odds ratio was as follows: 43.95 (95% CI = 36.61–52.78), with the error odds ratio as follows: 0.136 (95% CI = 0.155–0.119). The phi coefficient was 0.646 (95% CI = 0.632–0.658) and the RIOC was 0.883 (95% CI = 0.866–0.9).

The SN, SP, PPV, NPV, LR+, LR− and Fisher Exact P–values for individual studies are presented in Table 3. The cumulative SN was 70.97% (SD = 14.7) and the value ranged from 41.8% to 93.1%. The cumulative SP was 94.73% (SD = 5.8), and it ranged from 73% to 100%. The cumulative PPV was 0.9673 (SD = 0.036), and it ranged from 0.87 to 1. Cumulative NPV was 0.6324 (SD = 0.185) and it ranged from 0.33 to 0.96. The cumulative LR+ was 25.825 (SD = 39.72), and it ranged from 5.59 to 180.8. The cumulative LR– was 0.301 (SD = 0.145), and it ranged from 0.07 to 0.59.

### 3.5. Subgroup Analysis by IgM (Only) in the Acute Phase

The diagnostic odds ratio was as follows: 8.99 (95% CI = 7.248–11.157), with error odds ratio as follows: 0.051 (95% CI = 0.057–0.044). The phi coefficient was 0.389 (95% CI = 0.362–0.413) and the RIOC was 0.711 (95% CI = 0.693–0.73).

The SN, SP, PPV, NPV, LR+, LR− and Fisher Exact P–values for individual studies are presented in Table 4. The cumulative SN was 40.32% (SD = 26.02) and the value ranged from 11.7% to 89.9%. The cumulative SP was 93.01% (SD = 6.21) and it ranged from 83.9% to 100%. The cumulative PPV was 0.8074 (SD = 0.192), and it ranged from 0.39 to 1. The cumulative NPV was 0.6062 (SD = 0.205), and it ranged from 0.3 to 0.91. The cumulative LR+ was 6.487 (SD = 9.06), and it ranged from 0.85 to 28.56. The cumulative LR− was 0.6496 (SD = 0.293), and it ranged from 0.1 to 1.03.

### 3.6. Subgroup Analysis by NS1/IgM in the Acute Phase

The diagnostic odds ratio was as follows: 28.22 (95% CI = 24.179–32.946), with error odds ratio as follows: 0.479 (95% CI = 0.52–0.44). The phi coefficient was 0.637 (95% CI = 0.619–0.653) and the RIOC was 0.796 (95% CI = 0.776–0.815).

The SN, SP, PPV, NPV, LR+, LR− and Fisher Exact P–values for individual studies are presented in Table 5. For NS1/IgM, the cumulative SN was 78.62% (SD = 15.53), and the value ranged from 47.9% to 97.1%. The cumulative SP was 88.47% (SD = 12.44), and it ranged from 43.2% to 100%. The cumulative PPV was 0.9136 (SD = 0.11), and it ranged from 0.51 to 1. The cumulative NPV was 0.686 (SD = 0.168), and it ranged from 0.33 to 0.93. The cumulative LR+ was 8.08 (SD = 5.36), and it ranged from 1.55 to 26.03. The cumulative LR− was 0.24 (SD = 0.169), and it ranged from 0.03 to 0.56.

**Table 3 ijerph-19-08756-t003:** NS1 (Only) summary findings of post-hoc analysis.

No	Author, Year, Country (ref)	Region	Cohort Size	Prevalence (as Confirmed by Reference Method)	Reference Method	Sample Type	ICT Manufacturer	SN% (95% CI)	SP% (95% CI)	TP, FN, FP, TN	PPV (95% CI)	NPV (95% CI)	Positive Likelihood Ratio (+LR) (95% CI)	Negative Likelihood Ratio (−LR) (95% CI)	Fisher Exact *p*–Value
1	Kikuti, 2019, Brazil [25]	Americas	246	61.40%	NS1-ELISA, IgM-ELISA seroconversion (Abbott, Santa Clara, CA, USA; former Panbio Diagnostics, Brisbane, Australia), and/or RT-PCR	Acute serum	SD BIOLINE Dengue Duo RDT (Abbott, Santa Clara, CA, USA; former Alere Inc, Waltham, MA, USA)	NS1: 41.8% (35.1–48.7)	NS1: 98.0% (92.2–99.8)	61, 84, 2, 99	0.968 (0.886–0.994)	0.539 (0.511–0.548)	20.9 (5.409–121.666)	0.594 (0.573–0.664)	<0.0001 *
2a	Osorio, 2010, Colombia [26]	Americas	310	70.30%	RT-PCR, viral isolation and/or IgM seroconversion	Acute serum	Dengue NS1 Ag STRIP™ (Biorad Laboratories, Marnes–La–Coquette, France)	NS1: STRIP™ 61.5% (51.5–70.9)	NS1: STRIP™ 93.3% (84.2–99.4)	134, 84, 6, 86	0.956 (0.911–0.981)	0.506 (0.468–0.527)	9.179 (4.298–22.199)	0.413 (0.38–0.48)	<0.0001 *
2b	Osorio, 2010, Colombia [26]	Americas	310	70.30%	RT-PCR, viral isolation and/or IgM seroconversion	Acute serum	SD BIOLINE Dengue DUO^®^ (Standard Diagnostic Inc., Seoul, Korea)	SD Bioline™ 51.0% (44.1–57.7)	SD Bioline™ 96.7% (90.8–99.3)	111, 107, 3, 89	0.973 (0.924–0.993)	0.455 (0.426–0.466)	15.455 (5.139–59.676)	0.507 (0.484–0.57)	<0.0001 *
3a	Pal, 2014, Peru [27]	Americas	250	80%	RT-PCR and/or viral isolation followed by indirect immunofluorescence assay (IFA)	Acute serum	Dengue NS1 Ag STRIP^®^ (Bio–Rad, Marnes–La–Coquette, France)	NS1: Bio–Rad 79.1% (71.8–85.2)	NS1: Bio–Rad 100% (91.1–100.0)	158, 42, 0, 50	1 (0.974–1)	0.545 (0.5–0.545)	NE	0.209 (0.209–0.25)	<0.0001 *
3b	Pal, 2014, Peru [27]	Americas	250	80%	RT-PCR and/or viral isolation followed by indirect immunofluorescence assay (IFA)	Acute serum	Dengue NS1 Detect Rapid Test (1st generation) (InBios International, Seattle, WA, USA)	NS1: InBios 76.5% (64.6–85.9)	NS1: InBios 97.3% (86.2–99.9)	153, 47, 1, 49	0.991 (0.96–0.999)	0.509 (0.459–0.521)	28.333 (6.077–287.218)	0.242 (0.23–0.295)	<0.0001 *
3c	Pal, 2014, Peru [27]	Americas	250	80%	RT-PCR and/or viral isolation followed by indirect immunofluorescence assay (IFA)	Acute serum	Panbio Dengue Early Rapid	Panbio 71.9% (64.1–78.9)	Panbio 95.0% (83.1–99.4)	144, 56, 3, 47	0.983 (0.948–0.996)	0.458 (0.408–0.477)	14.38 (4.533–65.496)	0.296 (0.274–0.362)	<0.0001 *
3d	Pal, 2014, Peru [27]	Americas	250	80%	RT-PCR and/or viral isolation followed by indirect immunofluorescence assay (IFA)	Acute serum	SD Bioline Dengue NS1 Ag Rapid Test (Alere, Waltham, MA, USA)	SD 72.4% (64.5–79.3)	SD 100% (91.1–100)	145, 55, 0, 50	1 (0.971–1)	0.475 (0.436–0.475)	NE	0.276 (0.276–0.324)	<0.0001 *
4	Carter, 2015, Cambodia [28]	Asia	337	22.10%	Panbio Dengue IgM Combo ELISA (Panbio, Australia; Cat. # E-JED01C; Lot # 110061	Acute serum	SD BIOLINE Dengue DUO^®^ (Standard Diagnostic Inc., Seoul, Korea)	NS1: 60.8% (46.1–74.2)	NS1: 97.5% (94.9–99)	45, 29, 7, 256	0.873 (0.762–0.943)	0.898 (0.877–0.91)	24.32 (11.31–58.087)	0.402 (0.348–0.492)	<0.0001 *
5	Andries, 2012, Cambodia [29]	Asia	157	54.10%	NS1 capture ELISA, MAC-ELISA for IgM, indirect ELISA for IgG	Acute serum	SD BIOLINE Dengue DUO^®^ (Standard Diagnostic Inc., Seoul, Korea	NS1: 45.2% (36.4–54.3)	NS1: 96.8% (83.3–99.9)	38, 47, 2, 68	0.943 (0.825–0.988)	0.6 (0.558–0.616)	14.125 (3.989–71.882)	0.566 (0.53–0.671)	<0.0001 *
6a	Kulkarni, 2020, India [30]	Asia	809	38.60%	Panbio ELISA	Acute serum	J. Mitra Dengue Day 1 Test	NS1: J. Mitra–87.3 (82.2–92.5)	NS1: J. Mitra–93.4 (91.5–95.3)	272, 40, 33, 464	0.893 (0.862–0.917)	0.921 (0.903–0.936)	13.227 (9.968–17.687)	0.136 (0.108–0.171)	<0.0001 *
6b	Kulkarni, 2020, India [30]	Asia	809	38.60%	Panbio ELISA	Acute serum	SD–BIOLINE–Dengue–Duo (SDB–RDT)	SD–93.1 (88.2–98.0)	SD–97.8 (96.1–99.5)	291, 21, 11, 486	0.964 (0.941–0.979)	0.958 (0.944–0.967)	42.318 (25.505–74.766)	0.071 (0.055–0.094)	<0.0001 *
7	Vivek, 2017, India [31]	Asia	211	84.80%	RT-PCR	Acute serum	Dengue Day 1 Test (J. Mitra & Co)	NS1: 82.7% (76.3–87.9)	NS1: 96.9% (83.8–99.9)	148, 31, 1, 31	0.993 (0.964–1)	0.501 (0.431–0.516)	26.677 (4.823–520.66)	0.179 (0.168–0.237)	<0.0001 *
8	Sathish, 2003, India [32]	Asia	154	38.90%	NIV capture ELISA (MACELISA)	Acute serum	Panbio Rapid Immuochromatographic Card Test (Brisbane, Australia)	NS1: 73% (65–80)	NS1: 95% (90–98)	44, 16, 5, 89	0.903 (0.799–0.963)	0.847 (0.799–0.874)	14.6 (6.249–40.726)	0.284 (0.226–0.395)	<0.0001 *
9	Zainah, 2009, Malaysia [34]	Asia	314	31.80%	NS1 antigen-capture ELISA or RT-PCR	Acute serum	DENGUE NS1 Ag STRIP (Bio–Rad, Marnes–La–Coquette, France)	NS1: 90.4%	NS1: 99.5%	90, 10, 1, 213	0.988 (0.942–0.999)	0.957 (0.938–0.961)	180.8 (34.896–2934.041)	0.096 (0.086–0.142)	<0.0001 *
10a	Kyaw, 2019, Myanmar [35]	Asia	202	69.30%	DENV specific IgM capture ELISA or DENV RNA isolation	Acute serum	CareUs Dengue Combo, Korea	NS1: CareUs 72.1% (63.9–79.4)	NS1: CareUs 87.1% (76.1–94.3)	101, 39, 8.54	0.927 (0.871–0.964)	0.58 (0.515–0.624)	5.589 (2.992–11.727)	0.32 (0.267–0.417)	<0.0001 *
10b	Kyaw, 2019, Myanmar [35]	Asia	202	69.30%	DENV-specific IgM capture ELISA or DENV RNA isolation	Acute serum	Humasis Dengue Combo, Korea	NS1: Humasis 68.6% (60.2–76.1)	NS1: Humasis 90.3% (80.1–96.4)	96, 44, 6, 56	0.941 (0.884–0.975)	0.56 (0.502–0.595)	7.072 (3.376–17.18)	0.348 (0.302–0.439)	<0.0001 *
10c	Kyaw, 2019, Myanmar [35]	Asia	202	69.30%	DENV-specific IgM capture ELISA or DENV RNA isolation	Acute serum	Wondfo Dengue Combo, China	NS1: Wondfo 67.1% (58.7–74.8)	NS1: Wondfo 91.9% (82.2–97.3)	94, 46, 5, 57	0.949 (0.892–0.981)	0.553 (0.498–0.583)	8.284 (3.656–22.319)	0.358 (0.317–0.447)	<0.0001 *
11a	Jang, 2019, Myanmar [36]	Asia	220	49.50%	IgM/IgG ELISAs or qRT-PCR	Acute serum	Humasis Dengue Combo NS1, IgG/IgM (Humasis, Gyeonggi-do, Korea)	NS1: Humasis 63.3% (53.5–72.3)	NS1: 100%	69, 40, 0, 111	1 (0.94–1)	0.735 (0.708–0.735)	NE	0.367 (0.367–0.421)	<0.0001 *
11b	Jang, 2019, Myanmar [36]	Asia	220	49.50%	IgM/IgG ELISAs or qRT-PCR	Acute serum	SD Bioline Dengue Duo NS1 Ag and IgG/IgM (SD Bioline, Korea)	NS1: SD Bioline 48.6% (38.9–58.4)	NS1: 100%	53, 56, 0, 111	1 (0.921–1)	0.665 (0.64–0.665)	NE	0.514 (0.514–0.574)	<0.0001 *
11c	Jang, 2019, Myanmar [36]	Asia	220	49.50%	IgM/IgG ELISAs or qRT-PCR	Acute serum	CareUS Dengue Combo NS1 and IgM/IgG kits (WellsBio, Suwon, Korea)	NS1: CareUs 79.8% (71.1–86.9)	NS1: 100%	87, 22, 0, 111	1 (0.954–1)	0.835 (0.804–0.835)	NE	0.202 (0.202–0.248)	<0.0001 *
12a	Liu, 2021, Taiwan [37]	Asia	173	78.60%	qRT-PCR	Acute serum	Dengue NS1 Ag Strip (Bio–Rad, France)	NS1: SD 89.7%	NS1: SD 91.9%	122, 14, 3, 34	0.976 (0.939–0.993)	0.708 (0.612–0.754)	11.074 (4.201–41.603)	0.112 (0.089–0.172)	<0.0001 *
12b	Liu, 2021, Taiwan [37]	Asia	173	78.60%	qRT-PCR	Acute serum	Dengue Ag Rapid Test–Cassette (CTK Biotech, Inc., Powey, CA, USA)	NS1: Bio–Rad 85.3%	NS1: Bio–Rad 94.6%	116, 20, 2, 35	0.983 (0.945–0.997)	0.637 (0.554–0.667)	15.796 (4.641–90.748)	0.155 (0.136–0.219)	<0.0001 *
12c	Liu, 2021, Taiwan [37]	Asia	173	78.60%	qRT-PCR	Acute serum	SD Dengue Duo (Standard Diagnostics, Inc., Seoul, Korea)	NS1: CTK 89%	NS1: CTK 73%	121, 15, 3, 34	0.979 (0.942–0.995)	0.697 (0.605–0.737)	12.714 (4.434–54.867)	0.118 (0.097–0.178)	<0.0001 *
13a	Tricou, 2010, Vietnam [39]	Asia	292	83.90%	RT-PCR	Acute serum	Bio-Rad NS1 Ag Strip	NS1 Bio–Rad: 61.6 (55.2–67.8)	100%	151, 94, 0, 47	1 (0.972–1)	0.333 (0.303–0.333)	NE	0.384 (0.384–0.441)	<0.0001 *
13b	Tricou, 2010, Vietnam [39]	Asia	292	83.90%	RT-PCR	Acute serum	SD Dengue Duo (NS1/IgM/IgG) lateral flow rapid tests	NS1 SD: 62.4 (56.1–68.5)	100%	153, 92, 0, 47	1 (0.972–1)	0.338 (0.308–0.338)	NE	0.376 (0.376–0.432)	<0.0001 *

* Stands for statistically significant Fisher Exact *p*-value finding.

**Table 4 ijerph-19-08756-t004:** IgM (Only) Summary Findings of Post–Hoc Analysis.

No	Author, Year, Country (Ref)	Region	Cohort Size	Prevalence (as Confirmed by Reference Method)	Reference Method	Sample Type	ICT Manufacturer	SN% (95% CI)	SP% (95% CI)	TP, FN, FP, TN	PPV (95% CI)	NPV (95% CI)	Positive Likelihood Ratio (+LR) (95% CI)	Negative Likelihood Ratio (−LR) (95% CI)	Fisher Exact *p*–Value
1	Kikuti, 2019, Brazil [25]	Americas	246	61.40%	NS1-ELISA, IgM-ELISA seroconversion (Abbott, Santa Clara, CA, USA; former Panbio Diagnostics, Brisbane, Australia), and/or RT-PCR	Acute serum	SD BIOLINE Dengue Duo RDT (Abbott, Santa Clara, CA, USA; former Alere Inc, Waltham, MA, USA)	IgM: 11.7% (7.7–16.8)	IgM: 94.6% (87.5–98.3)	18, 133, 5, 90	0.775 (0.559–0.912)	0.402 (0.38–0.416)	2.167 (0.798–6.484)	0.933 (0.881–1.024)	0.105
2	Carter, 2015, Cambodia [28]	Asia	337	22.10%	Panbio Dengue IgM Combo ELISA (Panbio, Australia; Cat. # E-JED01C; Lot # 110061	Acute serum	SD BIOLINE Dengue DUO^®^ (Standard Diagnostic Inc., Seoul, Korea)	IgM: 32.7% (20.0–47.5)	IgM: 86.2% (81.5–90.0)	23, 48, 37, 229	0.387 (0.279–0.502)	0.828 (0.804–0.852)	2.37 (1.447–3.771)	0.781 (0.65–0.913)	<0.0001 *
3a	Kulkarni, 2020, India [30]	Asia	809	38.60%	Panbio ELISA	Acute serum	J. Mitra Dengue Day 1 Test	IgM: J. Mitra–22.5 (17.1–27.9)	IgM: J. Mitra–93.6 (91.6–95.6)	70, 242, 32, 465	0.688 (0.595–0.77)	0.658 (0.644–0.67)	3.516 (2.335–5.344)	0.828 (0.785–0.879)	<0.0001 *
3b	Kulkarni, 2020, India [30]	Asia	809	38.60%	Panbio ELISA	Acute serum	SD-BIOLINE-Dengue-Duo (SDB-RDT)	SD–34.4 (27.7–41.1)	SD 94.5 (91.2–97.8)	107, 205, 27, 470	0.797 (0.724–0.857)	0.696 (0.682–0.708)	6.255 (4.177–9.522)	0.694 (0.656–0.743)	<0.0001 *
4a	Kyaw, 2019, Myanmar [35]	Asia	202	69.30%	DENV specific IgM capture ELISA or DENV RNA isolation	Acute serum	CareUs Dengue Combo, Korea	IgM: CareUs 67.1% (58.7–74.8)	IgM: CareUs 83.9% (72.3–92.0)	94, 46, 10, 52	0.904 (0.843–0.948)	0.53 (0.466–0.577)	4.168 (2.382–8.01)	0.392 (0.325–0.507)	<0.0001 *
4b	Kyaw, 2019, Myanmar [35]	Asia	202	69.30%	DENV-specific IgM capture ELISA or DENV RNA isolation	Acute serum	Humasis Dengue Combo, Korea	IgM: Humasis 13.6% (8.4–20.4)	IgM: Humasis 83.9% (72.3–92.0)	19, 121, 10, 52	0.656 (0.474–0.809)	0.301 (0.27–0.326)	0.845 (0.399–1.871)	1.03 (0.914–1.196)	0.658
4c	Kyaw, 2019, Myanmar [35]	Asia	202	69.30%	DENV-specific IgM capture ELISA or DENV RNA isolation	Acute serum	Wondfo Dengue Combo, China	IgM: Wondfo 19.3% (13.1–26.8)	IgM: Wondfo 95.2% (86.5–98.9)	27, 113, 3, 59	0.901 (0.738–0.974)	0.343 (0.315–0.356)	4.021 (1.246–16.543)	0.848 (0.801–0.964)	0.006 *
5a	Jang, 2019, Myanmar [36]	Asia	220	49.50%	IgM/IgG ELISAs or qRT-PCR	Acute serum	Humasis Dengue Combo NS1, IgG/IgM (Humasis, Gyeonggi-do, Korea)	IgM: Humasis 51.4% (41.6–61.1)	IgM: Humasis 98.2 (91.5–99.9)	56, 53, 2, 109	0.966 (0.879–0.994)	0.673 (0.642–0.683)	28.556 (7.419–168.217)	0.495 (0.472–0.568)	<0.0001 *
5b	Jang, 2019, Myanmar [36]	Asia	220	49.50%	IgM/IgG ELISAs or qRT-PCR	Acute serum	SD Bioline Dengue Duo NS1 Ag and IgG/IgM (SD Bioline, Korea)	IgM: SD Bioline 60.6% (50.7–69.8)	IgM: SD Bioline: 100%	66, 43, 0, 111	1 (0.937–1)	0.721 (0.694–0.721)	NE	0.394 (0.394–0.449)	<0.0001 *
5c	Jang, 2019, Myanmar [36]	Asia	220	49.50%	IgM/IgG ELISAs or qRT-PCR	Acute serum	CareUS Dengue Combo NS1 and IgM/IgG kits (WellsBio, Suwon, Korea)	IgM: CareUs 89.9% (82.7–94.9)	IgM: CareUs 100%	98, 11, 0, 111	1 (0.961–1)	0.91 (0.879–0.91)	NE	0.101 (0.101–0.141)	<0.0001 *
6	Kittigul, 2002, Thailand [38]	Asia	92	56.50%	4x increased titers on hemagglutination inhibition test	1–6 days	Panbio Duo cassette IgM/IgG (Inverness, Australia)	IgM/IgG: 79%	IgM/IgG: 95%	41, 11, 2, 38	0.954 (0.857–0.992)	0.777 (0.692–0.81)	15.8 (4.601–91.658)	0.221 (0.18–0.343)	<0.0001 *

* Stands for statistically significant Fisher Exact *p*-value finding.

**Table 5 ijerph-19-08756-t005:** NS1/IgM Summary Findings of Post–Hoc Analysis.

No.	Author, Year, Country (Ref)	Region	Cohort Size (N)	Prevalence (as Confirmed by Reference Method)	Reference Method	Days Post Fever Onset	ICT Manufacturer	SN% (95% CI)	SP% (95% CI)	TP, FN, FP, TN	PPV (95% CI)	NPV (95% CI)	Positive Likelihood Ratio (+LR) (95% CI)	Negative Likelihood Ratio (−LR) (95% CI)	Fisher Exact *p*–Value
1	Kikuti, 2019, Brazil [25]	Americas	246	61.40%	NS1-ELISA, IgM-ELISA seroconversion (Abbott, Santa Clara, CA, USA; former Panbio Diagnostics, Brisbane, Australia), and/or RT–PCR	1–4 days	SD BIOLINE Dengue Duo RDT (Abbott, Santa Clara, CA, USA; former Alere Inc, Waltham, MA, USA)	NS1/IgM: 47.9% (41.0–54.8)	NS1/IgM: 92.6% (84.9–93.5)	72, 79, 7, 88	0.911 (0.832–0.96)	0.528 (0.49–0.551)	6.473 (3.116–19.921)	0.563 (0.513–0.655)	<0.0001 *
2a	Osorio, 2010, Colombia [26]	Americas	310	70.30%	RT-PCR, viral isolation, and/or IgM seroconversion	1–7 days	(i) Dengue NS1 Ag STRIP™ (Biorad Laboratories, Marnes–La–Coquette, France), (ii) SD BIOLINE Dengue DUO^®^ (Standard Diagnostic Inc., Korea)	NS1/IgM: SD Bioline™ 78.4% (72.4–83.7)	NS1/IgM: SD Bioline™ 91.3% (83.6–96.2)	171, 47, 8, 84	0.955 (0.919–0.978)	0.641 (0.591–0.672)	9.011 (4.784–18.911)	0.237 (0.206–0.292)	<0.0001 *
2b	Osorio, 2010, Colombia [26]	Americas	310	70.30%	RT–PCR, viral isolation, and/or IgM seroconversion	1–7 days	(i) Dengue NS1 Ag STRIP™ (Biorad Laboratories, Marnes–La–Coquette, France), (ii) SD BIOLINE Dengue DUO^®^ (Standard Diagnostic Inc., Seoul, Korea)	NS1/IgM/IgG: SD Bioline™ 80.7% (75–85.7)	NS1/IgM/IgG: SD Bioline™ 89.1% (81–94.7)	176, 42, 10, 82	0.946 (0.91–0.971)	0.661 (0.607–0.698)	7.404 (4.266–14.041)	0.217 (0.183–0.274)	<0.0001 *
3	Carter, 2015, Cambodia [28]	Asia	337	21.10%	Panbio Dengue IgM Combo ELISA (Panbio, Australia; Cat. # E-JED01C; Lot # 110061	1–2 days	SD BIOLINE Dengue DUO^®^ (Standard Diagnostic Inc., Seoul, Korea)	NS1/IgM: 57.8% (45.4–69.4)	NS1/IgM: 85.3% (80.3–89.5)	41, 30, 39, 227	0.511 (0.42–0.595)	0.884 (0.855–0.91)	3.932 (2.721–5.527)	0.495 (0.372–0.636)	<0.0001 *
4a	Kulkarni, 2020, India [30]	Asia	809	38.60%	Panbio ELISA	1–7 days	J. Mitra Dengue Day 1 Test	NS1/IgM: J. Mitra–58.3 (52.9–63.8)	NS1/IgM: J. Mitra–91.1 (88.6–93.6)	182, 130, 44, 453	0.804 (0.755–0.848)	0.777 (0.757–0.794)	6.551 (4.899–8.859)	0.458 (0.414–0.51)	<0.0001 *
4b	Kulkarni, 2020, India [30]	Asia	809	38.60%	Panbio ELISA	1–7 days	SD–BIOLINE–Dengue–Duo (SDB–RDT)	NS1/IgM: SD–55.7 (49.4–62.0)	NS1/IgM: SD–92.0 (87.5–96.5)	174, 138, 40, 457	0.814 (0.762–0.858)	0.768 (0.749–0.784)	6.963 (5.107–9.617)	0.482 (0.44–0.533)	<0.0001 *
5	Vivek, 2017, India [31]	Asia	211	84.80%	RT-PCR	1–5 days	Dengue Day 1 Test (J. Mitra & Co)	NS1/IgM: 89.4% (83.9–93.5)	NS1/IgM: 93.8% (79.2–99.2)	160, 19, 2, 30	0.987 (0.959–0.998)	0.612 (0.52–0.647)	13.545 (4.207–72.433)	0.113 (0.098–0.165)	<0.0001 *
6	Ngim, 2021, Malaysia [33]	Asia	368	45.40%	ELISA and/or RT-PCR	1–6 days	Dengue Combo Rapid Test–Cassette (Chembio Diagnostics, Inc., Medford, NY, USA)	NS1/IgM: 62.3%	NS1/IgM: 87.3%	104, 63, 26, 175	0.803 (0.737–0.858)	0.736 (0.7–0.766)	4.906 (3.378–7.283)	0.432 (0.368–0.515)	<0.0001 *
7a	Kyaw, 2019, Myanmar [35]	Asia	202	69.30%	DENV specific IgM capture ELISA or DENV RNA isolation	1–7 days	CareUs Dengue Combo, Korea	NS1/IgM: CareUs 92.1% (86.4–96.0)	NS1/IgM: CareUs 75.8 (63.3 –85.8)	129, 11, 15, 47	0.896 (0.856–0.926)	0.809 (0.711–0.883)	3.806 (2.631–5.505)	0.104 (0.058–0.18)	<0.0001 *
7b	Kyaw, 2019, Myanmar [35]	Asia	202	69.30%	DENV-specific IgM capture ELISA or DENV RNA isolation	1–7 days	Humasis Dengue Combo, Korea	NS1/IgM: Humasis 74.3% (66.2–88.2)	NS1/IgM: Humasis 88.7 (78.1– 95.3)	104, 36, 7, 55	0.937 (0.884–0.971)	0.605 (0.54–0.646)	6.575 (3.368–14.647)	0.29 (0.243–0.378)	<0.0001 *
7c	Kyaw, 2019, Myanmar [35]	Asia	202	69.30%	DENV-specific IgM capture ELISA or DENV RNA isolation	1–7 days	Wondfo Dengue Combo, China	NS1/IgM: Wondfo 70.0% (61.7–77.4)	NS1/IgM: Wondfo 91.9 (82.2 –97.3)	98, 42, 5, 57	0.951 (0.896–0.981)	0.576 (0.519–0.607)	8.642 (3.834–23.221)	0.326 (0.287–0.411)	<0.0001 *
8	Jang, 2019, Myanmar [36]	Asia	220	49.50%	IgM/IgG ELISAs or qRT-PCR	3–7 days	CareUS Dengue Combo NS1 and IgM/IgG kits (WellsBio, Suwon, Korea)	NS1/IgM: CareUs 96.3% (90.9–99.0)	NS1/IgM: CareUs 96.3% (90.9–98.9)	135, 5, 2, 60	0.983 (0.953–0.996)	0.92 (0.856–0.948)	26.027 (8.976–120.218)	0.038 (0.024–0.074)	<0.0001 *
9a	Liu, 2021, Taiwan [37]	Asia	173	78.60%	qRT-PCR	1–5 days	SD Dengue Duo (Standard Diagnostics, Inc., Seoul, Korea)	NS1/IgM: SD 95.6%	NS1/IgM: SD 89.2%	130, 6, 4, 33	0.970 (0.938–0.989)	0.847 (0.735–0.91)	8.852 (4.099–23.755)	0.049 (0.027–0.098)	<0.0001 *
9b	Liu, 2021, Taiwan [37]	Asia	173	78.60%	qRT-PCR	1–5 days	Dengue NS1 Ag Strip (Bio-Rad, Marnes–La–Coquette, France)	NS1/IgM: Bio–Rad 91.9%	NS1/IgM: Bio–Rad 91.9%	125, 11, 3, 34	0.977 (0.941–0.994)	0.755 (0.656–0.804)	11.346 (4.373–42.13)	0.088 (0.066–0.143)	<0.0001 *
9c	Liu, 2021, Taiwan [37]	Asia	173	78.60%	qRT-PCR	1–5 days	Dengue Ag Rapid Test-Cassette (CTK Biotech, Inc., San Diego, CA, USA)	NS1/IgM: CTK 95.6%	NS1/IgM: CTK 70.3%	130, 6, 11, 26	0.922 (0.888–0.945)	0.813 (0.664–0.913)	3.219 (2.164–4.649)	0.063 (0.026–0.138)	<0.0001 *
9d	Liu, 2021, Taiwan [37]	Asia	173	78.60%	qRT-PCR	1–5 days	Dengue Ag Rapid Test-Cassette (CTK Biotech, Inc., San Diego, CA, USA)	NS1/IgM/IgG: CTK 87.8%	NS1/IgM/IgG: CTK 43.2%	119, 17, 21, 16	0.85 (0.814–0.885)	0.491 (0.336–0.64)	1.546 (1.193–2.091)	0.282 (0.153–0.539)	<0.0001 *
9e	Liu, 2021, Taiwan [37]	Asia	173	78.60%	qRT-PCR	1–5 days	SD Dengue Duo (Standard Diagnostics, Inc., Seoul, Korea)	NS1/IgM/IgG: SD 97.1%	NS1/IgM/IgG: SD 86.5%	132, 4, 5, 32	0.964 (0.993–0.981)	0.89 (0.772–0.957)	7.193 (3.765–14.072)	0.034 (0.012–0.08)	<0.0001 *
10a	Tricou, 2010, Vietnam [39]	Asia	292	83.90%	RT-PCR	1–7 days	Bio-Rad NS1 Ag Strip	NS1/IgM Bio-Rad: 83.3% (72.1–91.4)	100%	204, 41, 0, 47	1 (0.98–1)	0.535 (0.488–0.535)	NE	0.167 (0.167–0.201)	<0.0001 *
10b	Tricou, 2010, Vietnam [39]	Asia	292	83.90%	RT-PCR	1–7 days	SD Dengue Duo (NS1/IgM/IgG) lateral flow rapid tests	NS1/IgM SD Duo: 75.5% (69.6–80.8)	100%	185, 60, 0, 47	1 (0.977–1)	0.439 (0.4–0.439)	NE	0.245 (0.245–0.288)	<0.0001 *
10c	Tricou, 2010, Vietnam [39]	Asia	292	83.90%	RT-PCR	1–7 days	Bio-Rad NS1 Ag Strip	NS1/IgM/IgG Bio-Rad: 61.6%;	100%	151, 94, 0, 47	1 (0.972–1)	0.333 (0.303–0.333)	NE	0.384 (0.384–0.441)	<0.0001 *
10d	Tricou, 2010, Vietnam [39]	Asia	292	83.90%	RT-PCR	1–7 days	SD Dengue Duo (NS1/IgM/IgG) lateral flow rapid tests	NS1/IgM/IgG SD Duo: 83.7% (78.4–88.1)	100%	205, 40, 0, 47	1 (0.98–1)	0.541 (0.493–0.541)	NE	0.163 (0.163–0.197)	<0.0001 *
11	Andries, 2012, Cambodia [29]	Asia	157	54.10%	NS1 capture ELISA, MAC-ELISA for IgM, indirect ELISA for IgG	1–7 days	SD BIOLINE Dengue DUO^®^ (Standard Diagnostic Inc., Seoul, Korea)	NS1/IgM/IgG: 94.4% (88.9–97.7)	NS1/IgM/IgG: 90.0% (73.5–97.9)	80, 5, 7, 65	0.918 (0.864–0.949)	0.932 (0.864–0.971)	9.44 (5.39–15.824)	0.062 (0.025–0.133)	<0.0001 *

* Stands for statistically significant Fisher Exact *p*-value finding.

### 3.7. Subgroup Analysis by ICT Manufacturer

Subgroup analysis was performed to determine the influence of multiple ICT assays on diagnostic accuracy and inter-study heterogeneity. Studies were grouped according to the ICT assay used to calculate heterogeneity and diagnostic accuracy results for sample verification. Heterogeneity trends (Cochran’s Q and χ^2^) were calculated for studies that used the following nine ICTs for NS1 antigen detection, including:

(1) Dengue NS1 Ag STRIP™ (Biorad Laboratories, Marnes-La-Coquette, France) (Cumulative SN: 90.58%, SP: 96.94%), (2) SD Bioline Dengue NS1 Ag Rapid Test (Alere, Waltham, MA, USA) (Cumulative SN: 62.7%, SP: 95.53%), (3) Dengue NS1 Detect Rapid Test (1st generation) (InBios International, Seattle, WA, USA) (Cumulative SN: 76.5%, SP: 97.3%), (4) Panbio™ Dengue Early Rapid (Cumulative SN: 72.45%, SP: 95%), (5) J. Mitra Dengue Day 1 Test (Cumulative SN: 85%, SP: 95.15%), (6) Dengue Ag Rapid Test-Cassette (CTK Biotech, Inc., San Diego, CA, USA) (Cumulative SN: 85.3%, SP: 94.6%, (7) CareUS Dengue Combo (WellsBio, Seoul, Korea) (Cumulative SN: 75.95%, SP: 93.55%), (8) Humasis Dengue Combo (Humasis, Gyeonggi-do, Korea) (Cumulative SN: 65.95%, SP: 95.15%), and (9) Wondfo Dengue Combo, China (Cumulative SN: 67.1%, SP: 91.9). The χ^2^ finding was 158.818 (df = 8), *p* < 0.001, and I^2^ = 95%.

For IgM antibody detection, the following five ICTs were used: (1) SD BIOLINE Dengue DUO^®^ (Standard Diagnostic Inc., Suwon, Korea) (Cumulative SN: 34.85%, SP: 93.83%), (2) J. Mitra Dengue Day 1 Test (Cumulative SN: 22.5%, SP: 93.6%), (3) CareUS Dengue Combo (WellsBio, Seoul, Korea) (Cumulative SN: 78.5%, SP: 92%), (4) Humasis Dengue Combo (Humasis, Gyeonggi-do, Korea) (Cumulative SN: 32.5%, SP: 91.1%), and (5) Wondfo Dengue Combo, China (Cumulative SN: 19.3%, SP: 95.2%). The χ^2^ finding was 21.698 (df = 5), *p* < 0.001, and I^2^ = 95.14%.

### 3.8. Diagnostic Accuracy by Reference Assay

A subgroup analysis was performed to determine the influence of multiple reference assays on study diagnostic accuracy. Studies were grouped according to the reference assay used for sample verification to calculate the diagnostic accuracy results. Studies used: (1) ELISA only, (2) RT-PCR only, (3) HAI only, (4) or multiple reference assays (two or more).

The SN, SP, DOR, and LR+ for the diagnostic tests were yielded for different referencing methods. Only ELISA reference standards (cumulative SN: 56.125%, SP: 94.35%) led to a DOR of 21.362 (95% CI = 17.08–26.741) and LR+ of 9.934 (95% CI = 8.186–12.112). Only RT–PCR reference standards (cumulative SN: 72.53%, SP: 93.87%) yielded a DOR of 40.432 (23.297–71.211) and LR+ of 11.832 (95% CI = 7.338–19.835). Only one study referenced the hemagglutination inhibition test; therefore, summary trends could not be computed. Finally, when using multiple reference assays (combinations of ELISA and RT–PCR) as standards (cumulative SN: 79.48%, SP: 92.45%), the DOR was 47.429 (95% CI = 34.771–64.813), and LR+ was 10.527 (95% CI = 8.365–13.393). While large heterogeneity was noted between the reference assay subgroups, the DOR and LR+ findings were the most notable with multiple reference assays, followed by RT-PCR, and finally ELISA.

### 3.9. Diagnostic Accuracy by Primary/Secondary Disease

Primary and secondary infection ICT interpretation (as defined by the manufacturer using IgM and IgG results) was compared with a valid reference assay to detect primary and secondary dengue infection. The overall sensitivity for primary infection was 75.68%, with SD = 25.45, whereas for secondary infection, the sensitivity was 71.9% (SD = 20.12). The dengue ICTs showed more sensitivity for primary infection.

### 3.10. Identification of Different DENV Serotypes

The sensitivity was different for DENV 1–4 serotypes between subgroups. The dengue ICTs gave the most sensitive results for DENV 3 (cumulative SN = 83.63%). DENV 1 was the second most sensitive test finding, with cumulative SN being 81.3%. Findings for DENV 2 were the third most sensitive (cumulative SN = 75.22%), followed by DENV 4 (62.06%).

### 3.11. Summary Receiver Operating Characteristics (SROC) Findings

The SROC curve analysis of NS1, NS1/IgM, and IgM study results are appended in Figure 5. The plot determined slightly improved (optimum SN and SP) results with NS1 alone, followed by NS1/IgM, and lastly, IgM alone. Therefore, the summary line that is closest to the upper-left corner of the plot (NS1 in this case) is considered nearest to the gold standard dengue testing format; this is closely followed by combined NS1/IgM testing, and lastly IgM, which is the furthest from the gold-standard metric.

## 4. Discussion

We conducted a systematic review and meta-analysis of NS1, and IgM and combined NS1/IgM antigen detection with different commercially-available ICTs to detect acute DENV infection. Five thousand two hundred two individuals were screened with a prevalence rate of 55.9% across 19 studies in dengue-endemic countries. We found that NS1 ICTs had more diagnostic potential (DOR: 48.35), followed by NS1/IgM ICTs (DOR: 27.87) and IgM ICTs (DOR: 10.54). NS1 ICTs had a higher pooled sensitivity (70.0%) and specificity (95.4%) in the acute phase compared with IgM ICTs (SN: 45.32%, SP: 92.71%) but lower than that of combined NS1/IgM ICTs (SN: 78.77%, SP: 88.25%). Nine NS1 ICTs had variable sensitivity (62.4–90.6%) and comparable specificity (93.6–92.9%). Similarly, variable sensitivity (19.3–78.5%) and comparable specificity (88.1–95.2%) was noted across six IgM ICT manufacturers. RT-PCR was the most accurate reference test, whereas ELISA was the least. The sensitivity of ICTs to detect primary infection was 77.9% compared with 66% for secondary infection in a subset of the studies that reported such data. The DENV-3 and DENV-1 serotypes were the most likely to be detected, compared with other serotypes. Overall, NS1 ICTs were the most predictive of acute DENV infection, with a higher detection in primary infection and DENV-3 serotypes. By different ICT manufacturers, the sensitivity varied, whereas the specificity was comparable. Our results emphasize the high diagnostic accuracy of NS1 ICTs in the acute phase, with certain commercial ICTs having higher sensitivities.

Our primary reason for conducting this study was to consider ICTs in detecting DENV in the acute phase to improve prompt diagnosis and early treatment. Many studies have demonstrated a high pooled sensitivity of NS1 ICTs and considerably low performance of IgM ICTs [14,15,16,17,18]. When used as a screening modality, ICTs may be part of the diagnostic algorithm that accounts for their high number of false negatives to optimize their performance [40]. While we did find a higher sensitivity of combined NS1/IgM ICTs, the overall diagnostic accuracy was prominently higher for NS1 ICTs. We only included acute-phase data, meaning only individuals presenting within seven days after the onset of symptoms were included. As we saw large inter-ICT manufacturer variability, we suggest considering ICTs with higher sensitivities/specificities for use in community settings, such as the NS1 Ag STRIP™ (Biorad Laboratories, Marnes-La-Coquette, France), similarly observed by Zhang et al. [16]. We considered these ICTs in the acute phase, since timely detection expands the diagnostic window of opportunity compared with gold standards (e.g., viral isolation, RT-PCR), which may take longer [41].

ICTs have the highest ability to distinguish serotypes DENV-3 and DENV-1, and the lowest for DENV-4, shown similarly by Zhang et al. and Shan et al. [16,17]. Among DENV-suspected individuals, NS1 ICTs and NS1/IgM ICTs are well-suited to detect acute infections of two serotypes, DENV-3 and DENV-1. Still, they are utilized at clinicians’ discretion to guide acute clinical management, as observed by Lim et al. [42]. Understanding different DENV serological sensitivity helps apply ICTs as DENV-3 serotypes are more widespread in certain regions (e.g., Sri Lanka, Malaysia, Vietnam) [43]. The global distribution of different dengue serotypes has implications for diagnostic strategies, and phylogeographic relationships with serotypes in specific regions may help guide the adaptation of ICTs [43].

It was possible to separately analyze primary and secondary dengue infection in 15 studies. All serum samples were collected between 0–7 days after symptom onset; patients with a primary infection were categorized as primary infection if negative for IgG, and a secondary infection if positive for IgG. Overall, a 77.9% sensitivity rate of NS1/IgM ICTs for primary infections was higher than that observed for IgM ICTs only (71%) by Blacksell et al. [14]; secondary infections had a 66% sensitivity rate, similarly observed by Blacksell et al. [14]. We were interested in identifying the performance of NS1 and IgM-based ICTs for the detection of primary and secondary infection in the acute phase. This was because secondary infections are more likely to lead to dengue hemorrhagic fever (DHF) and dengue shock syndrome (DSS), a more severe form of dengue fever (DF) [44]. Two hallmark symptoms of DHF are bleeding and plasma leakage due to increased vascular permeability and abnormal hemostasis [45]. A major loss of vascular fluids results in DSS, which puts the patients into hypovolemic shock and is associated with a mortality rate of higher than 50% when untreated, and is a medical emergency [46].

However, studies have demonstrated that DHF occurs in primary infections, which refutes the “original antigenic sin” hypothesis that individuals with secondary infections have a higher risk of DHF/DSS and the complex nature of DHF is also associated with age, sex, serotype, and genetic background [47]. Regardless, current data suggest that secondary dengue infections have a higher frequency of severe disease. A key feature of dengue ICTs as a satisfactory diagnostic test is discriminating between the two infection states [48]. The sensitivity ranged from 66–77.9%, which was less than optimal; however, we expect the specificity to be high in acute primary and secondary infections, as shown in a previous meta-analysis [14]. We could not detect specificity rates due to the scarcity of data in our studies. Regardless, we consider NS1/IgM ICTs to be a good test for differentiation of infection type in the acute phase to screen patients at risk for severe DENV infection sequelae. While outside the scope of this paper, the combined use of IgM and IgG has been shown to increase sensitivity during the first 7 days as IgG: IgM ratio > 1 is an excellent marker of secondary infection.

Our study has certain limitations. First, there was a lack of data on the adequate characterization of the samples by mean time since the onset of symptoms. Second, there was high variability in the performance of different ICTs; we did not conduct any further analyses to account for low-performing ICTs, but it is likely that if outlier ICTs were removed, the overall diagnostic potential of ICTs would be much higher. Third, we found that the reference assays had a high rate of heterogeneity compared with one another. We expect the ICTs to have variable efficacies due to different diagnostic reference standards. However, all studies used at least one reference test with high specificity, e.g., RT-PCRs, and ELISAs. Fourth, there was high heterogeneity within studies due to different methodologies. We, however, removed any studies that failed to meet the selection criteria. There may be different prevalence rates that may act as confounders for primary and secondary dengue infection burden. However, we considered dengue-endemic countries to eliminate the geographical bias of disease burden. The high heterogeneity may also be due to different control arms across the studies. While most studies screened suspected DF patients, certain studies used negative samples from dengue-endemic and non-dengue-endemic countries and patients diagnosed with other infections similar to DENV infection. Additionally, we cannot rule out the cross-reactivity of available RDTs with other flaviviruses, especially the Zika virus (ZIKV). However, Tan et al. reported through their surveillance data, that the cross-reactivity found in the lowest reactive titers of flaviviruses was generally higher than virus titers reported in natural infections due to respective flaviviruses. The cross-reactivity challenges can ideally be addressed through detection thresholds of assays designed for specific flaviviruses, e.g., ZIKV. Nevertheless, the cross-reactivity thresholds of commercially available assays are less likely to identify false positive results among clinical specimens in flavivirus-endemic regions. Last, due to incomplete data, we did not analyze for confounders such as age and gender.

Our study has a few strengths. We documented the entire search strategy and procedures; as such, we reported reasons for the removal of studies. We analyzed data for NS1, IgM, and combined NS1/IgM diagnostic accuracy, which provides insight into their ability to rule in and rule out disease. We obtained and analyzed data for different types of infection and by serology. Another strength of ours was the robust evaluation of many commercially-available ICTs, which had not been conducted previously. Overall, we found many methodological concerns in existing studies that examined the roles of ICT for DENV infection detection. It is important to adapt clear methodological guidelines for the assessment of DENV infection diagnostic tests such as the QUADAS-2 tool [22] and CASP checklist [49]. The dramatic increase in the dengue burden globally is alarming, and there is a need to regulate already available diagnostic tools for dengue across the healthcare sector [50].

## 5. Conclusions

In conclusion, we found that NS1 ICTs have good diagnostic accuracy and excellent pooled specificity for DENV detection in acute phases of infection within 7 days post–symptom onset. The NS1 ICTs can distinguish DENV-3 and DENV-1 more accurately. Type of infection, ICT manufacturers, reference methods, and DENV serotypes influence the diagnostic accuracy of these tests. There is a critical need to evaluate different ICTs for their diagnostic accuracy using standardized methodologies. Such data may be leveraged to incorporate ICTs as part of diagnostic algorithms in dengue-endemic regions with high burdens in these settings.

## Figures and Tables

**Figure 1 ijerph-19-08756-f001:**
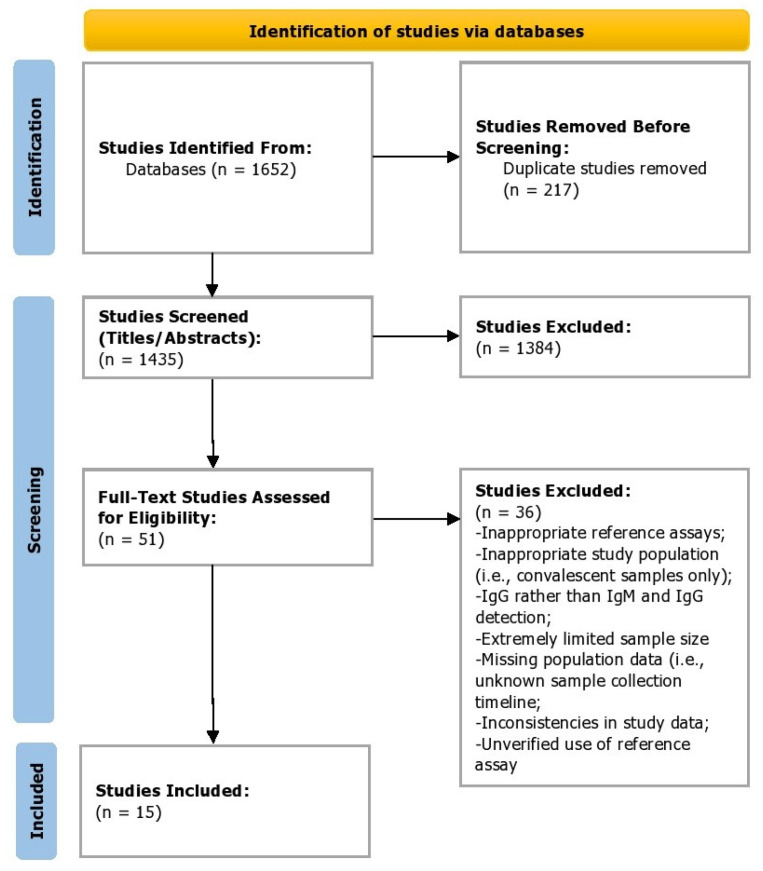
PRISMA flowchart of the study selection process.

**Figure 2 ijerph-19-08756-f002:**
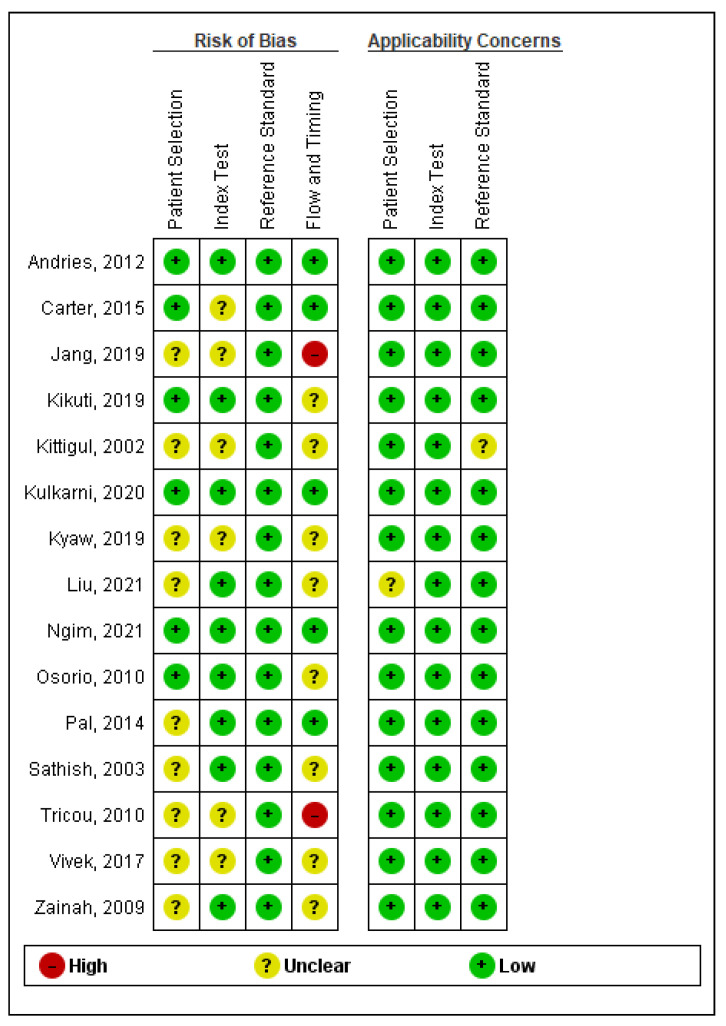
Risk of bias and applicability concerns summary: authors’ (ZS and AS) judgments about each domain for each included study [25,26,27,28,29,30,31,32,33,34,35,36,37,38,39].

**Figure 3 ijerph-19-08756-f003:**
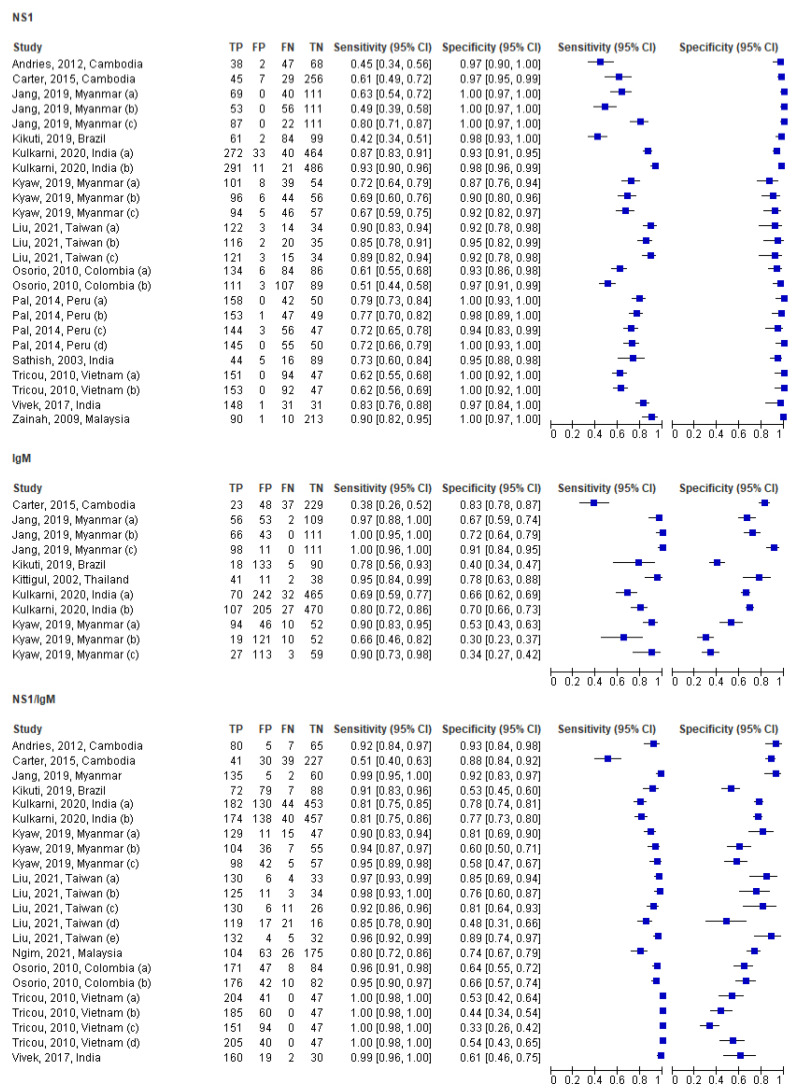
Summary forest plot for IgM (only), NS1 (only), and NS1/IgM tests in the acute phase [25,26,27,28,29,30,31,32,33,34,35,36,37,38,39].

**Figure 4 ijerph-19-08756-f004:**
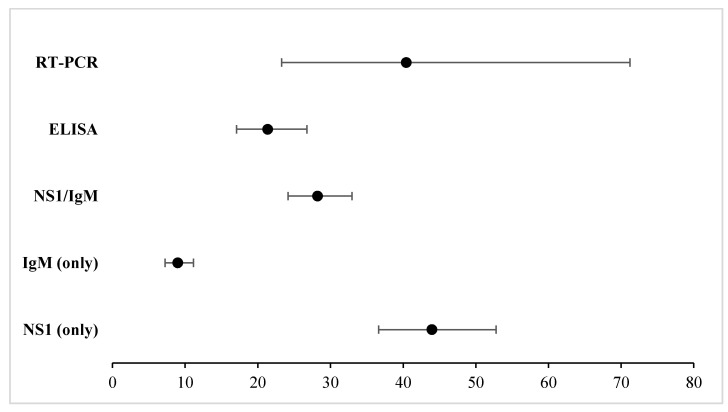
Summary findings of DOR when subgrouping ICTs: NS1 (only), IgM (only), NS1/IgM. Additionally, ELISA and RT-PCR were pooled to note reference test effects on DOR of DENV.

**Figure 5 ijerph-19-08756-f005:**
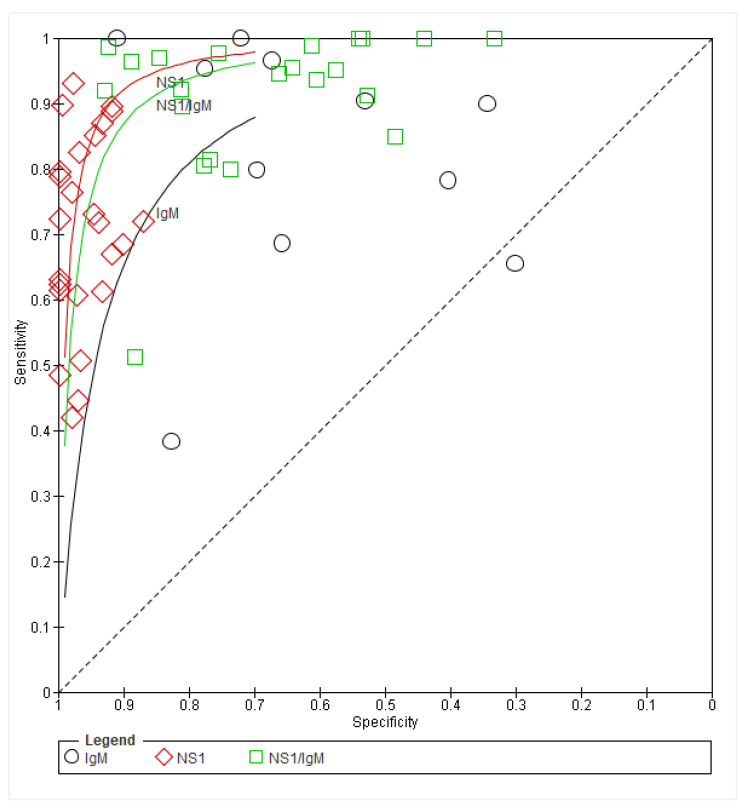
Summary ROC plot of tests: (1) IgM, (2) NS1, (3) NS1/IgM.

**Table 1 ijerph-19-08756-t001:** Definitions of listed statistical terminology.

Statistical Term	Definition
True Positives (TP)	Individuals with the Disease with the Value of the Parameter of Interest above the Cut-Off.
False Positive (FP)	Individuals without the disease with the value of the parameter of interest above the cut-off.
True Negative (TN)	Individuals without the disease with the value of the parameter of interest below the cut-off.
False Negative (FN)	Individuals with the disease with the value of the parameter of interest below the cut-off.
Positive Likelihood Ratio (LR+)	Measures how likely it is that a positive test result will occur in individuals with the disease compared with those without the disease.
Negative Likelihood Ratio (LR−)	Measures how likely it is that a negative test result will occur in individuals with the disease compared with those without the disease.
Positive Predictive Value (PPV)	Reports the proportions of positive diagnostic test results and the true positive results.
Negative Predictive Value (NPV)	Reports the proportions of negative diagnostic test results and the true negative results.
Diagnostic Odds Ratio (DOR)	A general estimate of the discriminative power of diagnostic procedures. It tests the ratio of positivity odds in individuals with disease related to the odds of individuals without the disease.
Error Odds Ratio (EOR)	Measures the likelihood of errors in diagnostic tests in individuals with the disease compared with those without.
Phi Coefficient	Also called a mean square contingency coefficient, this measures the association between two variables.
Relative Improvement Over Chance (RIOC)	This measures the predictive efficiency of the test.

**Table 2 ijerph-19-08756-t002:** (a) Characteristics of all the included studies. (b) Specific ICT manufacturers, sensitivity, and specificity in included primary studies.

(**a**)									
	**Author, Year, Country (Ref)**	**Region**	**DENV-Positive Individuals (n)**	**Cohort Size (N)**	**Prevalence (as confirmed by Reference Method)**	**Reference Method**		**Sample Type**	**Days Post Fever Onset**
1	Kikuti, 2019, Brazil [25]	Americas	151	246	61.40%	NS1-ELISA, IgM-ELISA seroconversion (Abbott, Santa Clara, CA, USA; former Panbio Diagnostics, Brisbane, Australia), and/or RT-PCR		Acute serum	1–4 days
2	Osorio, 2010, Colombia [26]	Americas	218	310	70.30%	RT-PCR, viral isolation, and/or IgM seroconversion		Acute serum	1–7 days
3	Pal, 2014, Peru [27]	Americas	200	250	80%	RT-PCR and/or viral isolation followed by indirect immunofluorescence assay (IFA)		Acute serum	1–5 days
4	Carter, 2015, Cambodia [28]	Asia	71	337	21.10%	Panbio Dengue IgM Combo ELISA (Panbio, Australia; Cat. # E-JED01C; Lot # 110061		Acute serum	1–2 days
5	Andries, 2012, Cambodia [29]	Asia	85	157	54.10%	NS1 capture ELISA, MAC-ELISA for IgM, indirect ELISA for IgG		Acute serum	1–7 days
6	Kulkarni, 2020, India [30]	Asia	312	809	38.60%	Panbio ELISA		Acute serum	1–7 days
7	Vivek, 2017, India [31]	Asia	179	211	84.80%	RT-PCR		Acute serum	1–5 days
8	Sathish, 2003, India [32]	Asia	60	154	38.90%	NIV capture ELISA (MACELISA)		Acute serum	2–7 days
9	Ngim, 2021, Malaysia [33]	Asia	167	368	45.40%	ELISA and/or RT-PCR		Acute serum	1–6 days
10	Zainah, 2009, Malaysia [34]	Asia	100	314	31.80%	NS1 antigen–capture ELISA or RT-PCR		Acute serum	1–7 days
11	Kyaw, 2019, Myanmar [35]	Asia	140	202	69.30%	DENV-specific IgM capture ELISA or DENV RNA isolation		Acute serum	1–7 days
12	Jang, 2019, Myanmar [36]	Asia	109	220	49.50%	IgM/IgG ELISAs or qRT-PCR		Acute serum	3–7 days
13	Liu, 2021, Taiwan [37]	Asia	136	173	78.60%	Qrt-PCR		Acute serum	1–5 days
14	Kittigul, 2002, Thailand [38]	Asia	52	92	56.50%	4× increased titers on hemagglutination inhibition test		Acute serum	1–6 days
15	Tricou, 2010, Vietnam [39]	Asia	245	292	83.90%	RT–PCR		Acute serum	1–7 days
(**b**)									
		**ICT Manufacturer**		**Sensitivity % (95% CI)**	**Specificity % (95% CI)**	**Combined Sensitivity % (95% CI)**	**Combined Specificity % (95% CI)**	**Primary and Secondary Sensitivitity (Acute) %**	
1	Kikuti, 2019, Brazil [25]	SD BIOLINE Dengue Duo RDT (Abbott, Santa Clara, CA, USA; former Alere Inc, Waltham, MA, USA)		NS1: 41.8% (35.1–48.7), IgM: 11.7% (7.7–16.8)	NS1: 98.0% (92.2–99.8), IgM: 94.6% (87.5–98.3)	NS1/IgM: 47.9% (41.0–54.8)	NS1/IgM: 92.6% (84.9–93.5)	Primary: NS1 26.7% (14.6–41.9), IgM 4.4% (0.5–15.2), NS1/IgM 31.1% (18.2–46.7), Secondary: NS1 40.7% (33.8–47.9), IgM 15.6% (10.8–21.4), NS1/IgM 49.8% (42.6–56.9)	
2	Osorio, 2010, Colombia [26]	(i) Dengue NS1 Ag STRIP™ (Biorad Laboratories, Marnes-La-Coquette, France), and (ii) SD BIOLINE Dengue DUO^®^ (Standard Diagnostic Inc., Suwon, Korea)		NS1: STRIP™ 61.5% (51.5–70.9), SD Bioline™ 51.0% (44.1–57.7)	NS1: STRIP™ 93.3% (84.2–99.4), SD Bioline™ 96.7% (90.8–99.3)	NS1/IgM: SD Bioline™ 78.4% (72.4–83.7), NS1/IgM/IgG: SD Bioline™ 80.7% (75–85.7)	NS1/IgM: SD Bioline™ 91.3% (83.6–96.2), NS1/IgM/IgG: SD Bioline™ 89.1% (81–94.7)	–	
3	Pal, 2014, Peru [27]	(i) Dengue NS1 Ag STRIP^®^ (Bio–Rad, Marnes-La-Coquette, France), (ii) Dengue NS1 Detect Rapid Test (1st generation) (InBios International, Seattle, WA, USA), (iii) Panbio Dengue Early Rapid, and (iv) SD Bioline Dengue NS1 Ag Rapid Test (Alere, Waltham, MA, USA)		NS1: Bio–Rad 79.1% (71.8–85.2), InBios 76.5% (64.6–85.9), Panbio 71.9% (64.1–78.9), SD 72.4% (64.5–79.3)	NS1: Bio–Rad 100% (91.1–100.0), InBios 97.3% (86.2–99.9), Panbio 95.0% (83.1–99.4), SD 100% (91.1–100)	–	–	–	
4	Carter, 2015, Cambodia [28]	SD BIOLINE Dengue DUO^®^ (Standard Diagnostic Inc., Suwon, Korea)		NS1: 60.8% (46.1–74.2); IgM: 32.7% (20.0–47.5)	NS1: 97.5% (94.9–99); IgM: 86.2% (81.5–90.0)	NS1/IgM: 57.8% (45.4–69.4)	NS1/IgM: 85.3% (80.3–89.5)	–	
5	Andries, 2012, Cambodia [29]	SD BIOLINE Dengue DUO^®^ (Standard Diagnostic Inc., Suwon, Korea)		NS1: 45.2% (36.4–54.3)	NS1: 96.8% (83.3–99.9)	NS1/IgM/IgG: 94.4% (88.9–97.7)	NS1/IgM/IgG: 90.0% (73.5–97.9)	Primary: NS1 89.5% (66.9–98.7), IgM/IgG 42.1% (20.3–66.5), NS1/IgM/IgG 100% (82.4–100), Secondary: NS1 43.4% (32.5–54.7), IgM/IgG 79.5% (69.2–87.6), NS1/IgM/IgG 97.6% (91.6–99.7)	
6	Kulkarni, 2020, India [30]	J. Mitra Dengue Day 1 Test and SD-BIOLINE-Dengue-Duo (SDB-RDT)		NS1: J. Mitra-87.3 (82.2–92.5), SD–93.1 (88.2–98.0); IgM: J. Mitra–22.5 (17.1–27.9), SD–34.4 (27.7–41.1)	NS1: J. Mitra–93.4 (91.5–95.3), SD–97.8 (96.1–99.5); IgM: J. Mitra–93.6 (91.6–95.6), SD–94.5 (91.2–97.8)	NS1/IgM: J. Mitra–58.3 (52.9–63.8), SD–55.7 (49.4–62.0)	NS1/IgM: J. Mitra–91.1 (88.6–93.6), SD–92.0 (87.5–96.5)	Sensitivity J. Mitra: NS1: Primary–67/72 (93.1%), Secondary–43/40 (>100%)/ IgM: Primary–14/65 (21.5%), Secondary–19/73 (26%)/ Combined NS1/IgM: Primary–72/113 (63.7%), Secondary–47/79 (59.5%)	
7	Vivek, 2017, India [31]	Dengue Day 1 Test (J. Mitra & Co)		NS1: 82.7% (76.3–87.9)	NS1: 96.9% (83.8–99.9)	NS1/IgM: 89.4% (83.9–93.5)	NS1/IgM: 93.8% (79.2–99.2)	Primary: NS1/IgM 90%, Secondary: NS1/IgM 96.9%	
8	Sathish, 2003, India [32]	Panbio Rapid Immuochromatographic Card Test (Brisbane, Australia)		NS1: 73% (65–80)	NS1: 95% (90–98)	–	–	–	
9	Ngim, 2021, Malaysia [33]	Dengue Combo Rapid Test-Cassette (Chembio Diagnostics, Inc., Medford, NY, USA)		–	–	NS1/IgM: 62.3%	NS1/IgM: 87.3%	–	
10	Zainah, 2009, Malaysia [34]	DENGUE NS1 Ag STRIP (Bio-Rad, Marnes–La–Coquette, France)		NS1: 90.4%	NS1: 99.5%	–	–	Primary: NS1 92.3%, Secondary: NS1 79.1%	
11	Kyaw, 2019, Myanmar [35]	(1) CareUs Dengue Combo, Korea, (2) Humasis Dengue Combo, Korea, and (3) Wondfo Dengue Combo, China		NS1: CareUs 72.1% (63.9–79.4), Humasis 68.6% (60.2–76.1), Wondfo 67.1% (58.7–74.8)/ IgM: CareUs 67.1% (58.7–74.8), Humasis 13.6% (8.4–20.4), Wondfo 19.3% (13.1–26.8)	NS1: CareUs 87.1% (76.1–94.3), Humasis 90.3% (80.1–96.4), Wondfo 91.9% (82.2–97.3), IgM: CareUs 83.9% (72.3–92.0), Humasis 83.9% (72.3–92.0), Wondfo 95.2% (86.5–98.9)	NS1/IgM: CareUs 92.1% (86.4–96.0), Humasis 74.3% (66.2–88.2), Wondfo 70.0% (61.7–77.4)	NS1/IgM: CareUs 75.8 (63.3–85.8), Humasis 88.7 (78.1–95.3), Wondfo 91.9 (82.2–97.3)	Primary: NS1/IgM CareUs 87.3% (77.3–94.0%), Humasis 85.0% (74.1–92.0%), Wondfo 83.1% (72.3–90.9%); Secondary: NS1/IgM CareUs 97.1 % (89.9–99.6%), Humasis 63.8% (51.3–75.0%), Wondfo 56.5% (44.0–68.4%)	
12	Jang, 2019, Myanmar [36]	(i) Humasis Dengue Combo NS1, IgG/IgM (Humasis, Geyonggi-do, Korea), (ii) SD Bioline Dengue Duo NS1 Ag and IgG/IgM (SD Bioline, Korea), and (iii) CareUS Dengue Combo NS1 and IgM/IgG kits (WellsBio, Seoul, Korea)		NS1: Humasis 63.3% (53.5–72.3), SD Bioline 48.6% (38.9–58.4), CareUs 79.8% (71.1–86.9), IgM: Humasis 51.4% (41.6–61.1), SD Bioline 60.6% (50.7–69.8), CareUs 89.9% (82.7–94.9), IgG: Humasis 72.5% (63.1–80.6), SD Bioline 78.0 (69.0–85.5), CareUs 82.6% (74.1–89.2)	NS1: 100%, IgM: Humasis 98.2 (91.5–99.9), SD Bioline and CareUs 100%, IgG; 95.2 (86.7–99.0), SD Bioline and CareUs 100%	NS1/IgM: Humasis 81.7% (73.1–88.4), SD Bioline 80.7% (72.1–87.7), CareUs 96.3% (90.9–99.0)	NS1/IgM: Humasis 98.2% (91.5–99.9), SD Bioline 80.7% (72.1–87.7), CareUs 96.3% (90.9–98.9)	Primary: NS1 Humasis 77.3% (54.63–92.2), SD Bioline 72.7% (49.8–89.3), CareUS 90.9% (70.8–98.9), IgM Humasis 68.2% (45.1–86.1), SD Bioline 72.7% (49.8–89.3), CareUS 86.4% (65.1–97.1), NS1/IgM Humasis 86.4% (65.1–97.1), SD Bioline 90.9% (70.8–98.9), CareUS 90.9% (70.8–98.9), Secondary: NS1 Humasis 59.8% (48.7–70.2), SD Bioline 42.5% (32.0–53.6), CareUS 77.0 (66.75–85.36), IgM Humasis 47.1% (36.3–58.1), SD Bioline 57.5% (46.41–68.0), CareUS 90.8% (82.7–96.0), NS1/IgM Humasis 80.5% (70.6–88.2), SD Bioline 78.2% (68.0–86.3), CareUS 97.7% (91.9–99.7)	
13	Liu, 2021, Taiwan [37]	Dengue NS1 Ag Strip (Bio-Rad, Marnes-La-Coquette, France), Dengue Ag Rapid Test-Cassette (CTK Biotech, Inc., San Diego, CA, USA) and SD Dengue Duo (Standard Diagnostics, Inc., Suwon, Korea)		NS1: SD 89.7%, Bio–Rad 85.3%, CTK 89%	NS1: SD 91.9%, Bio–Rad 94.6%, CTK 73%	NS1/IgM: SD 95.6%, Bio–Rad 91.9%, CTK 95.6%, NS1/IgM/IgG: SD 97.1%, CTK 87.8%	NS1/IgM: SD 89.2%, Bio–Rad 91.9%, CTK 70.3%, NS1/IgM/IgG: SD 86.5%, CTK 43.2%	Primary: SD NS1 95.7%, Bio–Rad NS1 90.4%, CTK NS1 93.9%, SD NS1/IgM/IgG 97.4%, CTK NS1/IgM 95.7%, CTK NS1/IgM/IgG 97.4%/ Secondary: SD NS1 60%, Bio–Rad NS1 50%, CTK NS1 60%, SD NS1/IgM 90%, SD NS1/IgM/IgG 100%, CTK NS1/IgM 90%, CTK NS1/IgM/IgG 100%	
14	Kittigul, 2002, Thailand [38]	Panbio Duo cassette IgM/IgG (Inverness, Australia)		–	–	IgM/IgG: 79%	IgM/IgG: 95%	Primary: IgM/IgG 67%, Secondary: IgM/IgG 80%	
15	Tricou, 2010, Vietnam [39]	Bio-Rad NS1 Ag Strip and SD Dengue Duo (NS1/IgM/IgG) lateral flow rapid tests		NS1 Bio-Rad: 61.6 (55.2–67.8); NS1 SD: 62.4 (56.1–68.5)	100%	NS1/IgM Bio–Rad: 83.3% (72.1–91.4); NS1/IgM/IgG Bio–Rad: 61.6%; NS1/IgM SD Duo: 75.5% (69.6–80.8); NS1/IgM/IgG SD Duo: 83.7% (78.4–88.1)	100%	Primary: NS1 Biorad 80.3% (68.7–89.1), SD NS1 80.3% (68.7–89.1), SD NS1/IgM 83.3% (72.1–91.4), SD NS1/IgM/IgG 83.3% (72.1–91.4), Secondary: NS1 Biorad 55.1% (47.4–62.6), SD NS1 56.3% (48.6–63.7), SD NS1/IgM 72.7% (65.5–79.2), SD NS1/IgM/IgG 84.1% (77.8–89.2)	

## Data Availability

The data presented in this study are available in the article.

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
