# Peer review of "Diagnostic Accuracy of Various Immunochromatographic Tests for NS1 Antigen and IgM Antibodies Detection in Acute Dengue Virus Infection"

_ijerph, 2022, doi:10.3390/ijerph19148756_

Round 1

Reviewer 1 Report

The study submitted by Haider et al is an extensive literature review that aims to highlight the potential of Rapid Diagnostic tests (RDT) for detection of Dengue virus (DENV) acute infections. For that, 1,652 studies were initially considered and 19 reliable databases were selected based on the use of commercial immunochromatographic serological tests (ICT) for the detection of DENV Non-Structural-1 protein (NS1), as well as DENV-Specific IgM. The performances of the tests were thoroughly assessed by different statistical methods and  parameters capable to affect the tests’ performances were evaluated, such as primary and secondary infections, correlation with the gold-standard test and  ability to detect the four dengue serotypes. The study reinforces the importance of tests based on detection of DENV NS1 protein and reinforces its use for the improvement of DENV diagnosis health authorities. In summary, the review represents a relevant contribution to the field and deserves publication. However, some suggestions are addressed to the authors aiming improvements of the present version of the manuscript, as indicated bellow:

Major points:

. The review shall take into account the potential cross reactive effects of NS1 expressed by other flavirusviruses in the performance of presently available tests, particularly regarding zika virus.

. It would be important to mention the reported sensitivity and specificity of the tests considered in the studies.

Minor points:

. More information regarding the statistical analyses shall be presented in order to help readers to understand the reported conclusions and discussions.

. The Table 1 could be divided in 2, since it shows totally different information separated by headlines.

. ELISA and RT-PCR data could also be reported, instead of only diagnostic odds ratio (DOR).

Author Response

Major points

Comment 1: The review shall take into account the potential cross reactive effects of NS1 expressed by other flavirusviruses in the performance of presently available tests, particularly regarding zika virus.

Response 1: The potential cross-reactivity of diagnostic assays has been discussed on lines 382-389.

Comment 2: It would be important to mention the reported sensitivity and specificity of the tests considered in the studies.

Response 2: The reported sensitivity and specificity has already been added in tables 2-4 for respective markers including NS1 and antibodies.

Minor points

Comment 3: More information regarding the statistical analyses shall be presented in order to help readers to understand the reported conclusions and discussions.

Response 3: A “Table 1” has been added with definitions of all listed statistical terms used throughout this paper. This operates as the glossary; in case any doubts arise.

Comment 4: The Table 1 could be divided in 2, since it shows totally different information separated by headlines. ELISA and RT-PCR data could also be reported, instead of only diagnostic odds ratio (DOR).

Response 4: Data on ELISA and RT-PCR has been added as prevalence which is classified as per a positive reference test. Diagnostic odds ratio (DOR) has not been reported in table 1 but in consequent tables. Table 1 has been divided into table 1a and 1b. Table 1a reports data of primary included studies including 1) author, year country, 2) region, 3) incidence, 4) cohort size, 5) prevalence confirmed by reference method, 6) reference method, 7) sample type, and 8) days post onset fever. Table 1b includes 1) ICT manufacturer, 2) sensitivity, 3) specificity, 4) combined sensitivity, 5) combined specificity, and 6) primary/secondary sensitivity.  

Reviewer 2 Report

1. The article number  11, 15 and 16 included  cases with post fever duration of more than 7 days also. In the discussion section  (lines 316-17, 334-337)  it has been mentioned  about acute phase data (within 7 days). Also in article 10 the the sample taken  post fever onset is "NS". Can you please describe/ clarify/ discuss it ?

2. Not all the studies included had utilized similar (standard)  reference test. How can we conclude the statistical test (sensitivity specificity and other) ?

3. In secondary dengue IgG appears from early days. Any comments/ discussion  we can conclude in secondary dengue?

4. Need some discussion on cross reactivity of these test to other flavi virus infection. 

Author Response

Comment 1: The article number 11, 15 and 16 included cases with post fever duration of more than 7 days also. In the discussion section (lines 316-17, 334-337) it has been mentioned about acute phase data (within 7 days). Also in article 10 the sample taken post fever onset is "NS". Can you please describe/ clarify/ discuss it?

Response 1: In response to your insightful comments, all authors have removed studies number 11,15 and 16 since only acute phase data is included in the study. In line with not including unclear data, study number 10 has been omitted as well, and all statistical analysis has been recomputed.

Comment 2: Not all the studies included had utilized similar (standard) reference test. How can we conclude the statistical test (sensitivity specificity and other)?

Response 2: The reference tests have been considered based on their recommendation by health policy makers are acceptable as gold standard and proxies. We do not expect that there will be wide discrepancy in between these reference tests. Regardless, this has already been addressed as a limitation and highlighted on lines 370-374.

Comment 3: In secondary dengue IgG appears from early days. Any comments/ discussion we can conclude in secondary dengue?

Response 3: Relevant discussion on lines 363-365 has been added

Comment 4: Need some discussion on cross reactivity of these test to other flavivirus infection.

Response 4: We have addressed this as a limitation and discussed the potential challenges and solutions related to cross-reactivity of flavivurses.

Reviewer 3 Report

They were 19 studies selected by Mughees Haider and colleagues to analyse the accuracy of diagnostic of acute dengue virus infections. QUADAS-2 has been used for quality assessment and diagnostic odds ratio (DOR) was an important statistical analysis reference.

Major comments:

1. None of the supplementary tables or figures are found.

2. Calculation of DOR was uncleared, but it seems an important index to support the conclusion.

3. SROC should be discussed.

4. Key analytical/statistical methods should be introduced in introduction.

Minor comments:

1. Line 115: specificity (SN)? Or SP?

2. Line 163-167: please check the citations and blanket.

3. Line 167: Eight different ICTs … but there are “nine” has been listed.

4. Line 173: Seven different ICTs… but there only “six” has been listed.

5. Title of table 1 is uncleared, not sure what is “all included studies”.

6. Seems like the Table 1 was split into 2 sections which is Author to Days post fever onset and ICT Manufacturer to Serotype sensitivity%. The arrangement of table should be improved.

7. A blanket with number found after country in table 1, it seems to be a reference for each study and been referenced by following tables. However, it hasn’t been mentioned in the writing.

8. The order of figure 2 should be aligned with table 1.  

Author Response

Major comments

Comment 1: None of the supplementary tables or figures are found.

Response 1: The supplementary material has been moved to the main text and is no longer attached under supplementary materials. However, supplementary table 1 has been attached to the submission.

Comment 2: Calculation of DOR was unclear, but it seems an important index to support the conclusion.

Response 2: Additional information has been provided in the methodology and is highlighted for your perusal.

Comment 3: SROC should be discussed.

Response 3: Relevant information has been added to the methods and under the results section; this is intended to only provide a visual summary of the information that had already been provided in earlier sections.

Comment 4: Key analytical/statistical methods should be introduced in introduction.

Response 4: Thank you for your insightful comment. The terms have been tabulated in the methods section; the purpose of doing so is to have an index glossary for the reader.

Minor comments

Comment 5: Line 115: specificity (SN)? Or SP?

Response 5:  It is SP and has been corrected.

Comment 6: Line 163-167: please check the citations and blanket.

Response 6: Formatting has been corrected.

Comment 7: Line 167: Eight different ICTs … but there are “nine” has been listed.

Response 7: There are nine ICTs and this has been corrected.

Comment 8: Line 173: Seven different ICTs… but there only “six” has been listed.

Response 8: There are now 5 ICTs and this has been corrected.

Comment 9: Title of table 1 is unclear, not sure what is “all included studies”.

Response 9: Table 1 has been divided into two sub tables and “all included studies” has been changed to “primary included studies.” These are studies that were analyzed for our analytical review.

Comment 10: Seems like the Table 1 was split into 2 sections which is Author to Days post fever onset and ICT Manufacturer to Serotype sensitivity %. The arrangement of table should be improved.

Response 10: This has been improved and the table has been split to two different sections.

Comment 11: A blanket with number found after country in table 1, it seems to be a reference for each study and been referenced by following tables. However, it hasn’t been mentioned in the writing.

Response 11: This has now been mentioned in writing as well.

Comment 12: The order of figure 2 should be aligned with table 1. 

Response 12: Figure 2 appears before table 1 and has been done to discuss the bias of the studies prior to the narrative synthesis.

Round 2

Reviewer 3 Report

Significant improvements have been made.

All my comments are well addressed.